# Polyphenols: Role in Modulating Immune Function and Obesity

**DOI:** 10.3390/biom14020221

**Published:** 2024-02-14

**Authors:** Md Abdullah Al Mamun, Ahmed Rakib, Mousumi Mandal, Santosh Kumar, Bhupesh Singla, Udai P. Singh

**Affiliations:** Department of Pharmaceutical Sciences, College of Pharmacy, The University of Tennessee Health Science Center, 881 Madison Avenue, Memphis, TN 38163, USA; mmamun3@uthsc.edu (M.A.A.M.); arakib@uthsc.edu (A.R.); mmandal1@uthsc.edu (M.M.); ksantosh@uthsc.edu (S.K.); bsingla@uthsc.edu (B.S.)

**Keywords:** polyphenols, obesity, metabolic disorders, gut health, immunomodulation

## Abstract

Polyphenols, long-used components of medicinal plants, have drawn great interest in recent years as potential therapeutic agents because of their safety, efficacy, and wide range of biological effects. Approximately 75% of the world’s population still use plant-based medicinal compounds, indicating the ongoing significance of phytochemicals for human health. This study emphasizes the growing body of research investigating the anti-adipogenic and anti-obesity functions of polyphenols. The functions of polyphenols, including phenylpropanoids, flavonoids, terpenoids, alkaloids, glycosides, and phenolic acids, are distinct due to changes in chemical diversity and structural characteristics. This review methodically investigates the mechanisms by which naturally occurring polyphenols mediate obesity and metabolic function in immunomodulation. To this end, hormonal control of hunger has the potential to inhibit pro-obesity enzymes such as pancreatic lipase, the promotion of energy expenditure, and the modulation of adipocytokine production. Specifically, polyphenols affect insulin, a hormone that is essential for regulating blood sugar, and they also play a role, in part, in a complex web of factors that affect the progression of obesity. This review also explores the immunomodulatory properties of polyphenols, providing insight into their ability to improve immune function and the effects of polyphenols on gut health, improving the number of commensal bacteria, cytokine production suppression, and immune cell mediation, including natural killer cells and macrophages. Taken together, continuous studies are required to understand the prudent and precise mechanisms underlying polyphenols’ therapeutic potential in obesity and immunomodulation. In the interim, this review emphasizes a holistic approach to health and promotes the consumption of a wide range of foods and drinks high in polyphenols. This review lays the groundwork for future developments, indicating that the components of polyphenols and their derivatives may provide the answer to urgent worldwide health issues. This compilation of the body of knowledge paves the way for future discoveries in the global treatment of pressing health concerns in obesity and metabolic diseases.

## 1. Introduction

Obesity is associated with an increased level of body fat mass caused by any specific or combination of contributing factors such as genetics, environmental factors, dietary habits, lifestyle, or multiple pathophysiological clinical conditions [1,2]. Obesity is a consequence of an energy imbalance between caloric input and output, and when this phenomenon continues for a long time, it results in metabolic disorders [3]. Obesity has been emerging as a major health concern worldwide in recent years, resulting in incalculable social costs [4]. Recently, the World Health Organization (WHO) declared obesity as an epidemic hazard worldwide. Currently, more than 1 billion people are affected by obesity and this prevalence is increasing day by day [5]. The multifactorial nature of obesity drives researchers to various treatment approaches, as the management of obesity is a slow process, including managing through physical exercise and diet control. Thus, medications, either natural or synthetic, are preferable to patients for effective outcomes during obesity [6]. There are several effective medications available nowadays, but even a few years back, phentermine and orlistat were the only drugs approved by the FDA as anti-obesity drugs [7]. Currently, orlistat, phentermine/topiramate, naltrexone/bupropion, and liraglutide are the anti-obesity drugs that have been FDA-approved for chronic weight management [8]. However, significant side effects of orlistat include gastrointestinal issues, liver damage, allergic reactions, and aberrant endocrine system responses shown during treatment [9]. Additionally, existing therapeutics for obesity are expensive, and some people may experience serious negative side effects.

Obesity has been shown to affect both innate and adaptive immune systems [10]. The emergence of numerous obesity-related issues brought about by altered innate and adaptive immune responses induces chronic adipose tissue (AT) inflammation [11]. Epidemiological data from various studies have demonstrated that obese people have a higher incidence and severity of different types of infections than lean participants [12]. However, the etiology of altered innate immune responses in obesity is still an intensely debated issue to date. Studying the relationship between immune response and metabolism has become more popular recently, and uncovering this role may shed light on the defective innate immunity during obesity. According to research on the subject of “immunometabolism”, immune cell function is linked to a certain state of cellular metabolism [13]. The significance of metabolism in enabling functional alterations to immune cells by blocking glycolysis in activated macrophages to stop inflammatory cytokine release has been shown [14]. Further, research has revealed that, when endogenous and external stimuli activate macrophages and other innate immune cells, these cells undergo unique metabolic rewiring for appropriate functional responses [15]. Thus, it is crucial to understand the mutual interaction that exists between immunity and obesity.

To this end, plant products have served as the most ancient source of medication from the start of civilization [16]. According to recent data, around three quarters of the total global population still rely on plant-derived therapeutic agents [17]. For example, many plants are used traditionally for treating diabetes, and the safety and efficacy of these plant-derived substances have been validated through many successful clinical trials. At present, the vast resources of phytochemicals are being explored intensely to unmask their antiadipogenic and anti-obesity properties [18], because phytochemicals are considered to be safe compared to synthetic compounds. Polyphenols are beneficial plant compounds with highly antioxidant properties [19]. Mounting evidence reports the role of polyphenolic compounds as being highly effective with lower side effects for the management of obesity-related complications. In this review, we will describe the mechanisms of how polyphenols are associated with ameliorating obesity conditions.

## 2. Diversity of Polyphenols and Their Effects

Approximately 8000 distinct polyphenols exist today, and they all have an aromatic ring with hydroxyl groups as part of their common phenolic structure. Despite being classified chemically as substances with phenolic structural characteristics, polyphenols are a wide class of natural products that include many subgroups of phenolic substances. Polyphenols can be found in abundance in fruits, vegetables, whole grains, and other forms of food and drink, including tea, chocolate, and wine [20]. Mounting evidence suggests that dietary polyphenols help to prevent obesity by influencing the brain’s different neurohormones that regulate satiety and food intake. Experimental studies indicate that polyphenols may play a part in the neurohormones that regulate energy balance and calorie intake in obese people [21]. Further, curcumin treatment resulted in the downregulation of the insulin-like growth factor (IGF) pathway in medulloblastoma cells [22]. Polyphenols can be categorized into various groups based on the number of aromatic rings and structural moiety connected to them [23]. The majority of them are produced by plants’ secondary metabolite shikimate pathway as a defense strategy [24]. There is enough evidence to conclude that the appropriate consumption of polyphenols has several positive health effects, even though most diseases are notlinked to inadequate or absent polyphenol intake [25]. Most of these advantageous traits of polyphenols are thought to be a result of their capacity to scavenge free radicals, create stable complexes, and obstruct subsequent chemical events. Additionally, they can scavenge hydrogen peroxide or limit its synthesis, protecting against oxidative stress, which might modulate immunological responses [26]. Depending on factors like the quantity of aromatic rings, their distribution in nature, and other factors, polyphenols can be categorized in many ways. Below is a basic description of the main groups of polyphenols.

### 2.1. Phenolic Acids

Benzoic acids and cinnamic acids and their derivatives are the two classes into which phenolic acids can be classified. Benzoic acids are the most basic phenolic acids found in nature, containing seven carbon atoms (C6–C1). While cinnamic acids can have nine carbon atoms (C6–C3), plants are the most prevalent source of these acids, which have seven. These compounds are identified by the presence of one or more hydroxyl and/or methoxyl groups, a carboxylic group, and a benzenic ring in the molecule [27]. While bound phenolic acids are connected to the cell walls, free phenolic acids are present in the pericarp, testa, and aleurone, the outer layers of the kernel. The majority of sorghum’s phenolic chemicals are found bound and in the bran. In sorghum, ferulic acid is the most prevalent bound phenolic acid; nevertheless, other, more abundant phenolic acids include syringic, protocatechuic, caffeic, p-coumaric, and sinapic [28]. Considering these different phenolic acids, for example, ferulic acid is reported to reduce the activity of hepatic lipogenic enzymes, including glucose-6-phosphate dehydrogenase (G6PD), malic enzyme (ME), and fatty acid synthase (FAS), which are in charge of synthesizing fatty acids and cholesterol [29]. Moreover, it has been reported that ferulic acid effectively reduces high fat diet (HFD)-induced visceral obesity and weight gain by modulating the peptide hormones associated with food regulation such as ghrelin, leptin, and insulin (Table 1) [30].

### 2.2. Flavonoids

Flavonoids are mostly studied and found in large quantities in foods and drinks that people consume daily, such as fruits, vegetables, tea, chocolate, and wine [39]. Flavonoids are found to reduce the risk associated with type 2 diabetes [40,41], cardiovascular diseases [42], obesity, and non-alcoholic fatty liver disease [43,44]. The beneficial effects of flavonoids have been thoroughly studied at the molecular level, with AMP-activated protein kinase (AMPK) activation being a common focal point. An essential function of AMPK is to regulate adipogenesis and lipid metabolism. Anabolic pathways like fatty acid production and gluconeogenesis are suppressed, while catabolic processes like fatty acid oxidation (FAO), glucose absorption, and glycolysis are encouraged through the phosphorylation and activation of AMPK [45]. A recent study showed that quercetin, a flavonoid, suppressed the key adipogenic factors, including CCAAT-enhancer-binding proteinα (C/EBPα), C/EBPβ, peroxisome proliferator activated receptor γ (PPARγ), and triglycerides (TG) synthesis enzymes, such as diacylglycerol acyltransferase-1, lipin1 (Table 1) [31]. In addition, quercetin is reported to inhibit the MAPK signaling factors, including extracellular signal-regulated kinases1/2 (ERK1/2) and c-Jun N-terminal kinase in adipocytes and macrophages [31]. Moreover, certain doses of quercetin have been shown to downregulate different inflammatory signaling pathways, including Nuclear factor kappa B (NF-κB) and cyclooxygenase-2 (COX-2) [32].

### 2.3. Tannins

Dietary tannic acid is a naturally occurring polyphenolic substance that is frequently present in plants and has long been associated with negative nutritional effects. Advances in structure identification and separation technologies have allowed researchers to isolate and identify thousands of tannin monomers from plants in the last few decades. Proanthocyanidins, gallotannins, and ellagitannins are widely reported tannins that have many pharmacological effects. A great deal of pharmacological research has also been carried out, including on anti-inflammatory, antioxidant, anti-aging, hypoglycemic, lipid-lowering, anticancer, antibacterial, and antiviral properties [46,47,48,49]. Tannins can interact with a variety of viral targets. Herpes simplex radiolabelled virus particles have shown that the antiviral properties of galloylated and hydrolyzable condensed tannins stem from their ability to impede virus adsorption [50]. Certain anti-nutritional characteristics of hydrolysable tannins have been demonstrated, particularly in the nutrition of monogastric animals by building complexes with proteins, carbs, and starches. When hydrolysable tannins were used in varying dosages, the degree of these beneficial and detrimental effects on digestion was seen; however, in animal testing, smaller doses did not significantly affect the digestibility of crude protein, organic matter, or ash [51]. The intake of hydrolysable tannins in higher amounts decreases weight gain and daily food intake [52,53]. In a recent study, You et al. reported that cyanidin-3-glucoside, a most abundant anthocyanin in plants, mitigates obesity by increasing energy expenditure and thermogenic properties in brown AT (BAT) (Table 1) [33].

### 2.4. Coumarins

Coumarins are molecules with both natural and synthetic origins that are among the most researched natural substances concerning human health. They belong to a family of heterocyclic compounds and have been thoroughly investigated in the domains of biochemistry and pharmacology [54]. As lipid-lowering agents, coumarins with various heterocycles based on the cyclization of 2-ethoxy-3-phenylpropanoic acid and 2-benzylmalonic acid have been studied to prevent the buildup of TGs and cholesterol in the walls of arteries or blood vessels, which causes atheroma and lowers the incidence of cardiovascular disease [55]. In a mouse model of hypercholesterolemia, the lipid-lowering properties of coumarin 7,8-dihydroxy-3-(4-methylphenyl) coumarin were evaluated, and it resulted in a considerable drop in serum cholesterol levels (Table 1) [34]. Reports showed that a HFD supplemented with 0·05% coumarin for eight weeks dramatically lowered blood lipid levels, body weight, and body fat [35]. The antiadipogenic properties of coumarins have also been demonstrated and found to suppress lipogenic gene expressions and lipid accumulation in 3T3-L1 adipocyte cells, potentially via the PPARγ pathway [56,57].

## 3. Polyphenols Alter Obesity

Anti-obesity phytochemicals are broadly classified into phenylpropanoids, flavonoids, terpenoids, alkaloids, glycosides, and phenolic acids according to their chemical structure [58]. Numerous studies have been conducted to determine the exact mechanisms associated with improving obesity and obesity-related complications using polyphenols. The inhibition of transcription factors involved in adipogenesis, such as C/EBP and PPAR, has been hypothesized as the method by which natural products alter obesity state [59]. One of the proposed mechanisms is to prevent the activation of C/EBP and PPAR signals, and the Wnt/β-catenin signaling pathway can prevent preadipocytes from differentiating into adipogenesis, which has an anti-obesity impact [60]. Several phytochemicals have been demonstrated to date in the literature to support uncoupled protein 1 (UCP1)-related adipose differentiation and thermogenesis capacity, boosting energy expenditure and reducing obesity and its side effects [61]. For a better understanding of the mechanism of polyphenols against obesity, a pictorial representation is shown in Figure 1.

### 3.1. Hormonal Regulation of Food Intake and Satiety

Mounting evidence suggests that dietary polyphenols prevent obesity by influencing the brain’s neurohormones that regulate satiety and food intake. Studies conducted in vitro and in vivo indicate that polyphenols may play a part in the neurohormones that regulate energy balance and calorie intake in obese people. Insulin is a vital hormone that controls blood sugar levels and cues for adipocytes to store energy. To date, research has not been able to adequately address the complicated and contentious relationships between insulin and obesity. However, the pathophysiology and development of obesity have been linked to the pancreas’ hypersecretion of insulin [21]. In the past, several studies have been conducted to determine how insulin affects the onset of obesity. For instance, a study found that, over a 15-year follow-up period, persons who hypersecreted insulin in response to an intravenous glucose tolerance test experienced abnormal weight gain [62]. Further, the effects of polyphenols on insulin highlight their potential significance in obesity, because insulin is a crucial neurohormone in the etiology of obesity. It has been shown that long-term resveratrol intracerebroventricular infusion alleviated hyperinsulinemia and corrected hyperglycemia in obese mice [63,64]. When curcumin was administered, the IGF pathway was found to be downregulated in medulloblastoma cells, and it has been shown that curcumin includes polyphenol metabolites that affect how the central nervous system regulates neurohormones like insulin [65]. Exciting studies performed in ob/ob and db/db mouse models of obesity led to this discovery, and profoundly influential studies in obese humans stimulated interest in leptin as a rational therapy for obesity and its numerous associated disorders. This leptin will forever be known as a potent regulator of feeding behavior and body weight [66]. Further, the daily intake of 200 mg/kg of resveratrol reestablished leptin sensitivity in obese rats and lowered their overall body weight [67]. Anthocyanins were reported to inhibit neuropeptide Y, reducing obesity in HFD-fed rats [68]. Polyphenols have also been proven to exert anti-obesity benefits by directly influencing neuropeptides in food intake [69].

### 3.2. Polyphenols Amend Pro-Obesity Enzymes

The primary enzyme responsible for breaking down dietary lipids in the digestive system is pancreatic lipase. Numerous clinical and experimental studies have demonstrated that lipase inhibitors can improve lipid metabolism in obese people. It has been shown that preventing the absorption of fatty acids lowers serum LDL (low-density lipoprotein) levels and raises HDL (high-density lipoprotein) levels [70]. Therefore, the creation of pancreatic lipase inhibitors is a crucial goal for the treatment of obesity. Pancreatic lipase activity has been inhibited both in vitro and in vivo by polyphenol-rich extracts from a variety of plants [71]. When fat synthesis outpaces fat oxidation, it leads to the development of obesity, and the body’s ability to store fat is closely controlled by lipogenesis and lipolysis. Lipolysis hydrolyzes TGs to produce free fatty acids, mono- or diacylglycerol, or free glycerol, whereas lipogenesis transforms simple sugars and other substrates into fatty acids and finally TGs. Thus, FAS is a crucial component of animal de novo lipogenesis. The processes in the body’s synthesis of endogenous lipids have the potential to become excessive and cause obesity. Towards this, polyphenols can disrupt the lipogenic pathway and inhibit the activity of the FAS enzyme to alter obesity. In HFD-induced obese rats, polyphenol treatment dramatically decreased the obesity index by preventing FAS activity [72]. Further, inhibiting FAS activity in 3T3-L1 cells, chokeberry-derived polyphenols drastically reduced body weight and blood TG levels [73]. A powerful natural FAS inhibitor, epigallocatechin-3-gallate has been shown to limit the enzyme’s activity in prostate cancer cells, lowering endogenous lipid production [74]. Taken together, uncovering better mechanisms of polyphenols in mediating obesity and metabolic syndrome has great potential to develop therapeutics for obesity.

### 3.3. Enhancement of Energy Expenditure through Polyphenols

Physical activity and thermogenesis are well-reported categories of total daily energy expenditure to date. Typically, an increase in TG lipolysis and FAO occurs simultaneously with the creation of heat. Hence, it will be helpful to stimulate thermogenesis for altering obesity conditions [75]. The two primary ATs in mammals, white AT (WAT) and BAT, are noticed. To control adiposity, BAT has a brown hue because of its strong vascularization and high mitochondrion concentration, which releases surplus energy through non-shivering thermogenesis. The inner membrane of the mitochondria uses UCP1 to assist BAT in using and dispersing the energy from lipids to produce heat [76]. Further, UCP1 is also reported to control BAT’s thermogenesis by reducing the proton gradient and separating oxidation from ATP production [77]. There is some distribution of WAT in the visceral region, such as in the omental, mesenteric, mediastinal, and epicardial regions, but the majority of WAT is stored in the subcutaneous region in the deep and superficial abdominal sections, as well as the gluteal-femoral regions [78]. WAT’s primary purpose is to store excess energy as TGs, which is counter to BAT’s ability to release energy by generating heat and warming the blood supply to essential organs [79]. Several animal studies have proven the role of polyphenolic compounds in the browning of AT, which is the key regulator of energy expenditure. By interfering with AMPK, sirtuin 1 (SIRT1), proliferator-activated-receptor-gamma-coactivator-1, catechol-O-methyltransferase, and sympathetic nervous system, which are key players in the transcriptional regulation and physiology of AT, phenolic compounds such as curcumin, quercetin, resveratrol, and isoflavones play crucial roles in thermogenesis and consequently lowering obesity [80]. The potential effect may be able to increase metabolic performance by efficiently controlling energy metabolism, as well as by boosting glucose uptake and mitochondrial activity, according to many preclinical studies [81]. Here, it becomes clear that polyphenolic substances such as gingerol, icariin, and resveratrol can affect skeletal muscle in preclinical metabolic syndrome models to control energy metabolism and enhance mitochondrial function [82]. In 3T3-L1 adipocytes, a polyphenolic extract of mango reduced adipogenesis and boosted thermogenesis, as demonstrated by increased expressions of thermogenic markers (UCP1, SIRT1, and AMPK) [83]. Expressions of C/EBPα and PPARγ, which are two important transcriptional factors for adipocyte differentiation, were also found to be suppressed in rodent models treated with polyphenolic extracts isolated from mango [84].

### 3.4. Regulation of Expression of Adipokines

Recent research has demonstrated that adipocytes have a secretory role and can release a variety of physiologic active compounds known as adipocytokines. Leptin and adiponectin are examples of adipokines that are selectively expressed in AT. Tumor necrosis factor (TNF)-α and interleukin-6 (IL-6) are examples of adipokines that are non-specifically expressed in AT. Propolis and its extract have been proven in studies to have an anti-obesity effect by altering adipokine secretion [85]. By influencing the hypothalamus’s weight regulation region, leptin reduces fat storage and prevents body weight gain by increasing energy intake and decreasing appetite. Brazilian green propolis was found to dramatically boost leptin expression when administered in vitro to the 3T3-L1 cell line and in vivo to C57BL/6 mice [86]. Adiponectin is another advantageous adipokine that controls an organism’s energy homeostasis, glucose metabolism, and fat metabolism [87]. An increase in adiponectin has been associated with enhanced AT and overall body energy metabolism [88]. Key metabolic processes are carried out by adiponectin in the liver and skeletal muscles. PPARα and AMPK are the mechanisms by which adiponectin’s effects on insulin sensitivity are mediated in muscle. Adiponectin stimulates FAO and reduces inflammation through the PPARα pathway, but it stimulates glucose transport and inhibits gluconeogenesis in the liver via AMPK [89]. Numerous plant-derived polyphenols, including resveratrol from red grapes, quercetin from fruits, vegetables, and grains, genistein from many plants, including soybeans, epigallocatechin gallate from green tea, berberine from *Coptis chinensis*, and curcumin from *Curcuma longa*, have been found to activate AMPK [90]. Since many of these substances are known to limit mitochondrial ATP generation, it appears that the mechanisms by which AMPK is activated necessitate the raising of AMP levels. Epigallocatechin-3-gallate, curcumin, resveratrol, and quercetin specifically target and block the mitochondrial F1F0-ATPase/ATP synthase [91,92]. Further evidence for the molecular basis of resveratrol and quercetin’s inability to activate AMPK in cells expressing the AMP-insensitive (R531G) AMPK2 subunit is provided by their failure to do so [93]. Taken together, it has been well established that polyphenols alter the levels of adipokines.

### 3.5. Regulation of Lipid Metabolism by Polyphenols

Numerous studies have investigated the role of polyphenols in regulating lipid metabolism, particularly in the context of obesity and cardiovascular health. Towards this, resveratrol, a polyphenol found in red wine, has been shown to affect lipid metabolism by enhancing the expression of genes involved in lipid oxidation and inhibiting lipogenesis, the process of fat storage [94,95]. Similarly, green tea catechins, such as epigallocatechin gallate, have demonstrated lipid-lowering effects by increasing the oxidation of fatty acids and reducing fat absorption in the intestines [96,97]. Further, quercetin, another polyphenol abundant in fruits and vegetables, has been reported to modulate lipid metabolism by reducing inflammation and oxidative stress, which are known contributors to dyslipidemia and atherosclerosis [98]. In addition, curcumin, a polyphenol from turmeric, has shown the potential to improve lipid profiles by increasing the expression of genes involved in cholesterol metabolism and reducing the formation of lipid plaques in blood vessels [99,100]. These studies collectively suggest that polyphenols can impact lipid metabolism through various mechanisms, including the regulation of gene expression, enhancement of lipid oxidation, inhibition of lipogenesis, and reduction in inflammation and oxidative stress. Thus, together, these effects have important implications for the prevention and management of conditions related to dyslipidemia, such as obesity and cardiovascular diseases.

## 4. Polyphenols Modulate Immune Response

### 4.1. Gut Health Reinforces the Immune Response

The increasing market size of immunity booster food and supplements indicates the concern of people worldwide regarding immunomodulation and protection from various diseases. It is always better to prevent infection and other disease conditions by boosting the natural immune system. Food contains relatively low levels of bioactive chemicals, yet their effects on health were constantly studied in the past. The mucosal layer, epithelium, and lamina propria are the three defense mechanisms of the intestinal innate immune system. The mucosal layer is the host’s initial line of defense against foreign pathogens in the intestinal tract [101]. Numerous investigations on the modulatory effects of polyphenols on intestinal immune function have produced compelling data that required more mechanistic studies. The nutritional protection of polyphenol-induced abnormal crypt lesions reduction may be a crucial step in the prevention of gastrointestinal tract tumors [102]. The bioactive substances known as polyphenols improve gut health by controlling mucosal immunity and inflammation. It has been demonstrated that polyphenols boost intestinal mucosal immunity in vivo after boosting the number of intraepithelial T cells and mucosal eosinophils in pigs infected with *Ascaris suum* [103]. The composition of the microflora populations may be modulated and subject to fluctuations by the phenolic substrates provided to the gut bacteria through varying dietary patterns and the aromatic metabolites generated, which have been shown to have selective prebiotic effects and antimicrobial activities against gut pathogenic bacteria [104,105]. Several studies have shown the critical function that gut bacteria plays in controlling the development of antigen-presenting cells [106]. The studies have shown that the monocolonization of germ-free (GF) mice with *Escherichia coli* was sufficient to recruit dendritic cells (DCs) to the intestines, and GF animals showed a decreased number of intestinal but not systemic DCs [107]. Furthermore, it has recently been demonstrated that microbe-derived ATP stimulates a subset of DCs that produce CD70 and CX3CR1 on their surface, which, in turn, causes T helper 17 (Th17) cells to differentiate [108]. Epigallocatechin-3-gallate, epicatechin-3-gallate, and epigallocatechin are examples of polyphenols that are said to increase IL-10 production by human white blood cells. Thus, they cause the activity of proinflammatory cytokines released by macrophages to decrease and increase the activity of anti-inflammatory cytokines [109]. The study targeted various types of immune cells, such as primary macrophages, to find out the potential targets [110]. Further, nitric oxide (NO) generation in healthy peripheral blood mononuclear cells (PBMC) was used as a model showing that red wine might cause human monocytes to produce NO and that the subsequently released NO’s vasodilatory properties could prevent atherosclerosis [111]. It also reduces the secretion of IL-6, TNF-α, and IL-1β from PBMCs [112]. Animal studies have shown that epigallocatechin-3-gallate reduces the signs and symptoms of autoimmune disorders. Mice given epigallocatechin-3-gallate had significantly more regulatory T (Treg) cells in their lymph nodes and spleens, and their T-cell response was reduced [113]. IL-10, transforming growth factor-beta-1 (TGF-β1), IL-6, and IL-17 are found to be balanced by polyphenols as a result of the regulation of Th17 and Tregs. Additionally, polyphenols block NF-κB activation, which prevents the development of dextran sulfate sodium-induced colitis [114]. A graphical representation of the roles of polyphenols in protecting gut health is shown in Figure 2.

### 4.2. Polyphenols Modulate Macrophage Functions and Inflammation

Macrophages are phagocytes that differentiate from transient monocytes, eliminate foreign substances, and initiate immune response. Macrophages like dendritic cells (DCs) serve as antigen-presenting cells, activating immature T cells into effector T cells in the presence of an antigen [115]. Macrophages are crucial for tissue healing, host defense, and controlling inflammation, and also play a harmful role in several chronic disorders, such as rheumatoid arthritis, inflammatory bowel disease, asthma, and atherosclerosis. The traditional classification of macrophages includes the traditional inflammatory M1 and immunosuppressive M2 phenotypes. The stimulation of interferon (IFN) and the activation of toll-like receptors (TLRs) by bacterial lipopolysaccharides (LPS) cause M1 differentiation to begin, whereas IL-4 causes M2 polarization to begin [116]. Certain polyphenols from cinnamon are reported to activate macrophages and have an impact on lowering inflammation and enhancing immunological performance [117,118]. Reactive oxygen species (ROS) govern a wide range of complex biological activities, including angiogenesis, inflammation, differentiation, and proliferation. The nicotinamide adenine dinucleotide phosphate (NADPH) oxidase family consists of seven members with tissue- and cell-type-specific expression profiles. The primary purpose of all family members is to produce controlled levels of ROS [119]. Flavanols can inhibit transcription factors (such as NF-κB), resulting in a decrease in NADPH oxidase activity [120]. The polarization of macrophages and the role of polyphenols are shown graphically in Figure 3.

In colitis, dietary polyphenols were able to encourage the phenotypic conversion of M1 into M2 macrophages and suppress colitis [121]. It has been shown that punicalagin and ellagic acid from pomegranate peel decreased TLR4 mRNA and protein expression levels in a dose-dependent manner, and inhibited LPS-induced intracellular ROS generation. Further, the inhibition of LPS-induced phosphorylation and the nuclear translocation of p65 were also facilitated by the anti-inflammatory mechanism [122]. In macrophage cell lines J774, it has been shown that pomegranate polyphenols dose-dependently reduced the macrophage response to M1 proinflammatory activation [123]. Accordingly, studies conducted both in vivo and in vitro have shown that one of the primary effects of polyphenols on macrophages is the inhibition of important inflammatory response regulators, with the most consistent effect being the repression of COX-2, inducible nitric oxide synthase (iNOS), and the cytokines TNF-α, IL-1β, and IL-6 [124]. When Chinese propolis was added to mouse RAW 264.7 macrophages challenged with LPS, a dose-dependent reduction in iNOS, IL-1β, and IL-6 mRNA expression was seen, followed by an increase in NO, IL-1β, and IL-6 production [125]. Some studies have described the implication of MAPK pathways for cell signaling implicated in these outcomes. The flavonoid procyanidin C1 was able to suppress the expression of TLR4 and COX-2, as well as phosphorylate p38 and ERK-1/2, and reduce the secretion of TNF-α, IL-1β, and IL-6 in bone-marrow-derived macrophages [126].

### 4.3. Modifying the Function of Natural Killer Cells by Polyphenols

The use of cytotoxic immune cells in the prevention and treatment of cancer is a growing field of research known as immune-cell-mediated cancer therapy. Natural killer (NK) cells, which also attack other microorganisms and alter body cells, are one of the promising cells for combating cancer. About 10–15% of blood lymphocytes are NK cells, also referred to as cytotoxic lymphocytes of the innate immune system. NK cells detect ‘stressed’ cells like tumor- or virus-infected cells, and then they eliminate those cells on their accord [127]. The impact of different plant-derived compounds on the ability of NK cells to fight against malignant illnesses has long been the subject of much research. Numerous fruits and vegetables naturally contain flavonoids, which are plentiful phytonutrients, and the subgroup called quercetin significantly affects cytotoxic immune cells [128]. Both endogenous and exogenous immune-modulating substances can either strengthen or weaken immunological responses and inflammation. Many secondary metabolites from plants, such as flavonoids like quercetin, have stimulating effects on NK cells and increase the ability of NK cells to destroy YAC-1 target cells [129]. Further, it has been shown that resveratrol modulates direct and indirect effects on NK cell function, which is consistent with the immune-modulating abilities of nutrition-derived substances [130]. By altering the expression of activating cell surface receptors such as NKG2D on NK cells or by stimulating the production of their corresponding ligands on cancerous cells, resveratrol appears to improve immune responses [131]. When viewed in light of the additional chemical characteristics of this natural substance, this immune modulation is intriguing. In distinct human hepatoblastoma cells, resveratrol was found to have an inhibitory effect on the classical histone deacetylases (class I, II, and IV). It has been shown that HDAC inhibition caused a dose-dependent decrease in the growth of cancer cells [132]. Further, the enhanced expression of various NKG2D ligands made leukemia K562 and gastric cancer SNU1 and SNU-C4 cells more susceptible to NK cell-mediated lysis [133]. Increased tumor resistance to treatment, whether it be chemotherapy, radiation, or any of the targeted therapies, is one of the most significant unsolved issues in cancer therapy. As a general rule, defects in apoptosis have a direct impact on this enhanced resistance of cancer therapy, and the solution to this issue might be autophagy, a different type of cell death. Numerous research studies have shown that polyphenolic substances, including rottlerin, genistein, quercetin, curcumin, and resveratrol, can mediate both canonical and non-canonical autophagy via multiple pathways and may offer novel therapeutic approaches in the treatment of cancer to address the serious issue of drug resistance in cancer therapy [134].

## 5. Role of Polyphenols in Obesity-Associated Immunomodulation

Obesity, which is characterized by the intricate activation of different inflammatory pathways, is crucial for contributing to insulin resistance, as well as other metabolic dysfunctions. Importantly, AT comprises both immune cells from the innate and adaptive immune systems. Immune cells from the innate and adaptive immune systems crosstalk with adipocytes during obesity-related inflammation. This crosstalk between adipocytes and other immune cells results in a vicious cycle of inflammatory responses, which causes more recruitment of immune cells such as monocytes/macrophages, neutrophils, and T cells, which exacerbates the obesity condition [135]. Therefore, therapeutic agents that can suppress the inflammatory behaviors of different immune cells may have the potential to ameliorate obesity and associated metabolic diseases.

It has been well established that polyphenols possess several pharmacological activities, including anti-inflammatory, antioxidant, and anticancer activities. As obesity-induced inflammation potentially increases the proinflammatory mediators and accumulation of immune cells, polyphenolic compounds might have therapeutic potential for treating obesity-related inflammatory conditions. It has been shown that curcumin suppresses the migration of macrophages in the mesenteric AT and significantly reduces the secretion of monocyte chemoattractant protein-1 (MCP-1) from RAW264.7 macrophages that are treated with the conditioned media from mesenteric AT [136]. In addition, curcumin supplementation has been found to reduce the expression of several pro-inflammatory M1 macrophage markers like CD11c, CD38, and CD80 and decrease adipocyte size compared to HFD-fed obese mice, as well as causing reduced macrophage infiltration in AT [36]. Moreover, curcumin, along with another polyphenolic compound, resveratrol, has been shown to inhibit NF-κB activation in adipocytes [37]. To this end, it has been shown that resveratrol suppresses the phosphorylation of the p65 subunit of NF-κB signaling in adipocyte cell lines, which is also well reported in several inflammatory signaling by canonical NF-κB signaling pathways [137]. Resveratrol also increases the accumulation of Tregs, as well as maintains glucose homeostasis by regulating SIRT1 signaling in HFD-fed mice [38]. Furthermore, capsaicin inhibits the p65 subunit of the NF-κB in adipocytes, as well as the mesenteric AT-conditioned medium and MCP-1-induced macrophage migration. Additionally, capsaicin significantly suppresses the activation of macrophages to produce inflammatory mediators like NO, TNF-α, and MCP-1, which are induced by the obese-mouse mesenteric AT-conditioned medium [138]. The polyphenol butein, which is an active component of *Toxicodendron vernicifluum*, is also reported to inhibit the NF-κB signaling activated by TNF-α in adipocytes, and butein prevents inflammation in the suppression of the NF-κB and MAPK signaling pathway in macrophages. [139]. The upregulation of pro-inflammatory genes, such as iNOS, is associated with the activation of MAPKs in macrophages [140]. 6-gingerol, available from ginger, has been reported to reduce the expression of adipocyte differentiation genes, such as C/EBPα,and PPARγ after HFD feeding. Moreover, animals fed an HFD along with 6-gingerol showed a reduced expression of the macrophage marker F4/80 in their WAT compared to mice fed an HFD alone, indicating a possible immunoregulatory role for 6-gingerol in the WAT [141]. Another polyphenolic compound from ginger, gingerenone A, was also reported to suppress adipocyte differentiation markers and the chemotaxis of macrophages, and, interestingly, promotes M2 macrophages in HFD-fed mice [142].

Further, polyphenols obtained from different plant extracts showed a notable improvement in ameliorating obesity conditions with inflammation. Polyphenol-rich fractions obtained from table grapes have significantly decreased the body weight and inflammatory gene expression in visceral fat in obese mice fed on an HFD [143]. Polyphenol-rich grape powder supplementation increases the glucose tolerance in HFD-fed mice compared to mice fed only with an HFD. Importantly, quercetin 3-O-glucoside, a polyphenol obtained from grape powder, decreased the expression of inflammatory markers like TNF-α, IL-6, and CD11c in the serum and AT of HFD-fed mice and MCP-1 and IL-1β in human primary adipocytes [144]. Moreover, polyphenols from tea suppressed obesity-related fatty liver and inflammatory cytokine expression in HFD-fed dogs [145]. These promising studies suggest that polyphenols have a beneficial effect in the prevention of AT inflammation under obese conditions.

## 6. Conclusions

In conclusion, polyphenols play a crucial role in immunomodulation and the fight against obesity, and have a lot of potential for enhancing human health. The health benefits of polyphenols present in a wide range of foods and beverages have been the subject of detailed research. The mechanisms by which polyphenols combat obesity are multifaceted, but the neurohormones in the brain that control insulin linked to hunger and fullness are the most studied field. Polyphenols such as resveratrol and curcumin have been demonstrated in animal models to reduce hyperinsulinemia and correct hyperglycemia, reducing inflammation and cancer, suggesting that they may be useful in the treatment of obesity. Furthermore, polyphenols can inhibit the lipogenic pathway and pro-obesity enzymes like pancreatic lipase, which lowers the synthesis and storage of fat. Polyphenols also increase thermogenesis, which increases energy expenditure and aids in weight control and calorie burning. The hormones leptin and adiponectin control the expression of adipocytokines, which are essential for maintaining energy homeostasis and metabolic processes. Polyphenols also influence mucosal immunity and inflammation to mediate the gut immune system and overall health. Polyphenols have been found to impact immune cells, such as T cells, macrophages, and NK cells. Polyphenols promote the production of anti-inflammatory cytokines while inhibiting proinflammatory cytokines, enhancing NK cell function, and making them valuable in preventing inflammation-related diseases. Taken together, these studies highlight the possibility that polyphenols support general health by reducing obesity and altering the immune system and inflammation. However, further studies are required to understand the precise mechanisms of how polyphenols, as well as their derivatives, enhance the immune system, reduce AT inflammation, and manage obesity, by utilizing the potential of some of these and developing tailored therapeutics in humans.

## Figures and Tables

**Figure 1 biomolecules-14-00221-f001:**
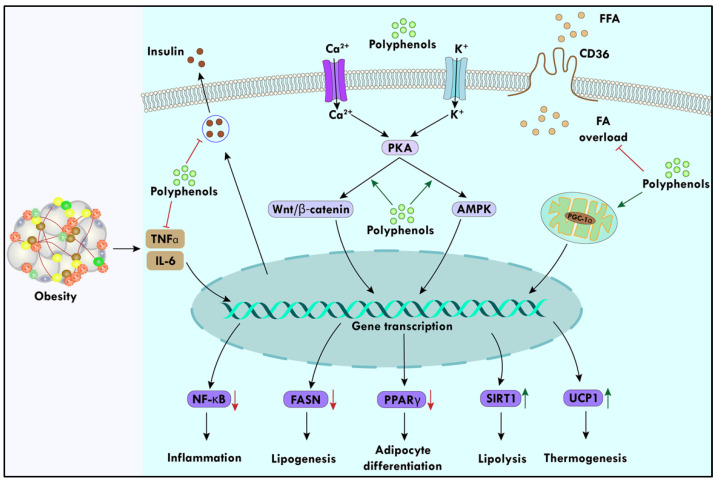
Polyphenols play a role in modifying obesity. They hinder the production of free fatty acids and mitigate the hypersecretion of insulin associated with obesity. Natural polyphenols facilitate BAT thermogenesis and lipolysis, aiding in the regulation of adiposity by enhancing the expression of UCP1 and SIRT1. Furthermore, polyphenols downregulate gene transcription of FAS and PPARγ, thereby inhibiting lipogenesis and adipocyte differentiation. Additionally, they prevent TNF-α and IL-6-mediated NF- κB expression, subsequently averting inflammation.

**Figure 2 biomolecules-14-00221-f002:**
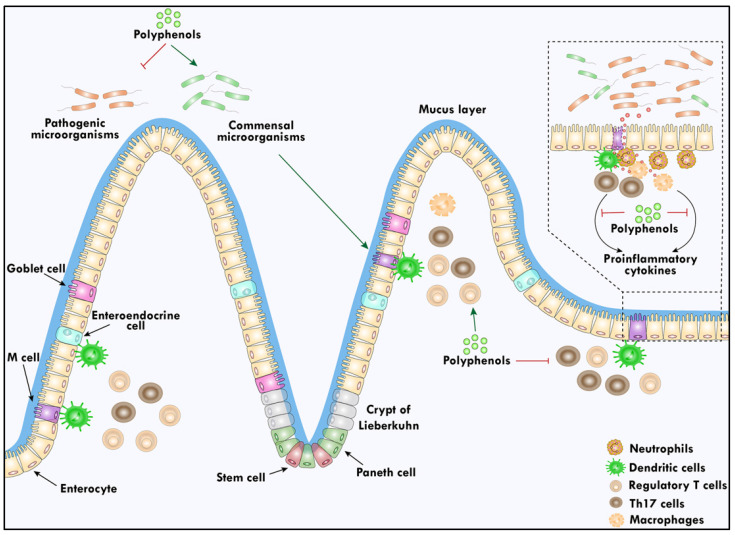
Polyphenols enhance the well-being of the digestive system by promoting a healthy community of beneficial bacteria and preserving an equilibrium between Th17 and Treg. Harmful microorganisms can harm the mucosal lining of the gut, allowing the invasion of antigens and triggering the activation of antigen-presenting cells, such as dendritic cells. Polyphenols intervene in this process, obstructing the pathway and averting subsequent inflammation driven by proinflammatory cytokines such as IL-10, IL-6, and IL-17.

**Figure 3 biomolecules-14-00221-f003:**
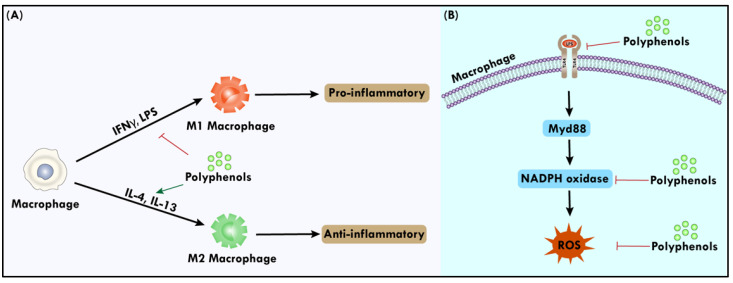
(**A**) The initiation of M1 differentiation is prompted by the activation of toll-like receptors (TLRs) and the stimulation of IFN through bacterial LPS. Conversely, M2 polarization is initiated by IL-4 and IL-13. Polyphenols promote the conversion of macrophages to anti-inflammatory M2 phenotype. (**B**) The generation of ROS is facilitated by the NADPH oxidase family. Polyphenols, particularly flavonols, can reduce NADPH activity, leading to a subsequent reduction in ROS production and inflammation.

**Table 1 biomolecules-14-00221-t001:** Summary of polyphenolic compounds and their functions in obesity.

Compound	Structure	Function	References
Ferulic acid	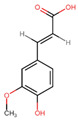	Reduces lipogenic enzyme activity Prevents HFD-induced visceral adiposity	[29,30]
Quercetin	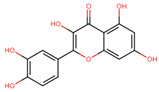	Suppresses adipogenic factors Reduces TG synthesis enzymes	[31,32]
Cyanidin-3-glucoside	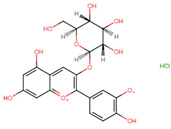	Increase energy expenditure Increase thermogenesis	[33]
7,8-dihydroxy-3-(4-methylphenyl) coumarin	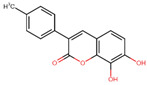	Lower serum cholesterol	[34,35]
Curcumin	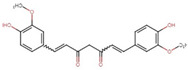	Decrease adipocyte size Reduce M1 macrophage marker	[36]
Resveratrol	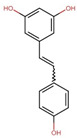	Inhibit NF-κB activation Increase Tregs	[37,38]

## Data Availability

No new data were created or analyzed in this study. Data sharing is not applicable in the review article.

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
