# Peer review of "Polyphenols: Role in Modulating Immune Function and Obesity"

_biomolecules, 2024, doi:10.3390/biom14020221_

Round 1

Reviewer 1 Report

Comments and Suggestions for Authors

I found the reading of the paper very interesting and well-structured, especially with the inclusion of biological activity graphs. In this regard, I would suggest adding a summary diagram of phenolic and flavonoid molecules in Section 2.

Review the formatting of the bibliography with corresponding line spacing.

Correctly cite the bibliography in the text using [] e not ()

Author Response

We thank all 2 reviewers for their time, and helpful comments and are encouraged by their enthusiasm for this review. We modified the review based on the reviewer’s suggestion.

Reviewer #1: I found the paper reading very interesting and well-structured, especially with the inclusion of biological activity graphs. In this regard, I suggest adding a summary diagram of phenolic and flavonoid molecules in Section 2.

Response: We agree with the reviewer’s thoughtful comment. We added a new summary diagram in the revised review.

  1. ii) Review the formatting of the bibliography with corresponding line spacing.

Response: We have modified the formatting of the bibliography with corresponding line spacing as suggested by the reviewers.

iii) Correctly cite the bibliography in the text using [] e not ()

Response: We agree with the reviewer’s comment. We have modified the bibliography in the text as suggested by the reviewers.

Reviewer 2 Report

Comments and Suggestions for Authors

In this review, the primary objective was to analyze the etiology of the immune response and its alterations in the context of obesity. Additionally, the aim was to explore how polyphenolic compounds can potentially impact the modulation of obesity-related complications through an immunomodulatory lens.

Major Issues:

While acknowledging the topic's significance, it is essential to note that the review needs to be revised to delve deeply into the specific subject of immunomodulation. The provided details remain relatively general, offering limited insight into the pathways through which polyphenols may confer immunomodulatory benefits in the context of obesity. Despite the description of various polyphenols, there is a notable absence of integration between obesity, immunomodulation, and polyphenols, failing to explore the signaling pathways already documented in the literature. Consequently, it seems unnecessary to elaborate on the properties of polyphenols and their relevance to obesity without discussing their specific effects on immunomodulation.

Miner issues:

 The review's title may not accurately represent its content.

On page 1, line 3, medicinal plants are described as a practice that people commonly misconceive; it might be more appropriate to mention that 75% of the population utilizes them. 

Pg. 1, Line 6: These are not properties of polyphenols.

Pg. 1, Line 9: This review methodically investigates the mechanism by which naturally occurring polyphenols mediate obesity and metabolic function in immunomodulation. 

Pg. 3, Line 9: Several references do not match the information. For example, "Long-term resveratrol intracerebroventricular infusion alleviated hyperinsulinemia and corrected hyperglycemia in obese mice (21)." 

Pg. 6, Line 6:  Same as above: “It has been shown that long-term resveratrol intracerebroventricular infusion alleviated hyperinsulinemia and corrected hyperglycemia in obese mice” (56).

Author Response

Reviewer #2: In this review, the primary objective was to analyze the etiology of the immune response and its alterations in the context of obesity. Additionally, the aim was to explore how polyphenolic compounds can potentially impact the modulation of obesity-related complications through an immunomodulatory lens.

Major Issues:

  1. i) While acknowledging the topic's significance, it is essential to note that the review needs to be revised to delve deeply into the specific subject of immunomodulation. The provided details remain relatively general, offering limited insight into the pathways through which polyphenols may confer immunomodulatory benefits in the context of obesity. Despite the description of various polyphenols, there is a notable absence of integration between obesity, immunomodulation, and polyphenols, failing to explore the signaling pathways already documented in the literature. Consequently, it seems unnecessary to elaborate on the properties of polyphenols and their relevance to obesity without discussing their specific effects on immunomodulation.

Response: We like to thank the reviewers and agree with the reviewer’s thoughtful comments and this will help us a lot to revise this review. In the revision we did; i) remove unnecessary descriptions of polyphenols, ii) integrate obesity and immunomodulation, iii) add signaling pathways, and, iv) added relevance to obesity. We also added a summary in the revised version.

Miner issues:

  1. ii)  The review's title may not accurately represent its content.

Response: We agree with the reviewer’s comments. However, based on the suggestion we added immunomodulation and integrated it with obesity. Now I feel this title reflects the core issue of the review and is accurate as described.

iii) On page 1, line 3, medicinal plants are described as a practice that people commonly misconceive; it might be more appropriate to mention that 75% of the population utilizes them. 

Response: We agree with the reviewer’s comments and modified this in the revised version.

  1. iv) Pg. 1, Line 6: These are not properties of polyphenols.

Response: We agree and modified this in the revised version.

  1. v) Pg. 1, Line 9: This review methodically investigates the mechanism by which naturally occurring polyphenols mediate obesity and metabolic function in immunomodulation. 

Response: We agree with the reviewer’s comments and modified this in the revised version.

  1. vi) Pg. 3, Line 9: Several references do not match the information. For example, "Long-term resveratrol intracerebroventricular infusion alleviated hyperinsulinemia and corrected hyperglycemia in obese mice (21)." 

Response: We agree with the reviewer’s comments and modified this in the revised version.

vii) Pg. 6, Line 6:  Same as above: “It has been shown that long-term resveratrol intracerebroventricular infusion alleviated hyperinsulinemia and corrected hyperglycemia in obese mice” (56).

Response: We agree with the reviewer’s comments and modified this in the revised version.